# Anti-Inflammatory and Antimicrobial Effect of Ellagic Acid and Punicalagin in Dermal Fibroblasts

**DOI:** 10.3390/ijms26178681

**Published:** 2025-09-05

**Authors:** Javier Ramos-Torrecillas, Anabel González-Acedo, Lucía Melguizo-Rodríguez, Concepción Ruiz, Elvira De Luna-Bertos, Rebeca Illescas-Montes, Olga García-Martínez

**Affiliations:** 1Biomedical Group (BIO277), Department of Nursing, Faculty of Health Sciences, University of Granada, Avda. Ilustración 60, 18016 Granada, Spain; jrt@ugr.es (J.R.-T.); luciamr@ugr.es (L.M.-R.); crr@ugr.es (C.R.); elviradlb@ugr.es (E.D.L.-B.); ogm@ugr.es (O.G.-M.); 2Institute of Biosanitary Research, ibs.Granada, C/Doctor Azpitarte 4, 18012 Granada, Spain; anabelglez@ugr.es; 3Biomedical Group (BIO277), Department of Nursing, Faculty of Health Sciences of Melilla, University of Granada, C/Santander 1, 52005 Melilla, Spain; 4Institute of Neuroscience, Centro de Investigación Biomédica (CIBM), University of Granada, Parque de Tecnológico de la Salud (PTS), Avda. del Conocimiento S/N, Armilla, 18016 Granada, Spain

**Keywords:** ellagic acid, punicalagin, fibroblasts, anti-inflammatory, anti-bacterial, chronic wound

## Abstract

Chronic wounds are characterized by persistent inflammation and microbial colonization, which interfere with the healing process and represent a significant clinical challenge. This study evaluated the anti-inflammatory and reparative potential of ellagic acid and punicalagin, along with their antimicrobial activity. Human dermal fibroblasts were exposed to a simulated inflammatory microenvironment induced by interleukin-1β (IL-1β), Interleukin-6 (IL-6), and Tumor Necrosis Factor-α (TNF) or bacterial lipopolysaccharides (LPS) and subsequently treated with ellagic acid or punicalagin (10^−6^ M and 10^−7^ M). Cell proliferation was assessed via MTT assay, and migration was evaluated using the scratch wound assay. IL-1β and IL-6 secretion was quantified by Enzyme-Linked Immunosorbent Assay in LPS-treated fibroblasts. Antimicrobial activity against *Candida albicans*, *Staphylococcus aureus*, and *Pseudomonas aeruginosa* was analyzed using the disk diffusion method. Both compounds significantly enhanced fibroblast viability and migration under inflammation and reduced the secretion of IL-1β and IL-6. However, no antimicrobial activity was observed at the tested concentrations. These findings suggest that ellagic acid and punicalagin may promote wound healing by modulating inflammation and supporting fibroblast function, despite lacking direct antimicrobial effect. Further in vivo studies are needed to validate their therapeutic relevance and explore their potential in the development of novel treatments for chronic wounds.

## 1. Introduction

Wound healing is a complex biological process that occurs in four interconnected phases—namely, hemostasis, inflammation, proliferation, and remodeling. These phases rely on the coordinated activation of distinct cellular populations that regulate tissue growth, repair, and remodeling, thereby restoring both structural and functional integrity [1,2].

The balance between inflammation and regeneration is critical for effective wound healing, with the inflammatory phase constituting a key component of the process [3]. During this phase, immune system activation initiates a controlled pro-inflammatory response that promotes pathogen clearance and triggers tissue repair signaling. However, prolonged or excessive inflammation can disrupt the healing cascade, delay tissue repair, and increase the risk of complications such as fibrosis, keloids, or chronic wounds. The latter pose a major clinical challenge, particularly in patients with comorbidities such as diabetes or vascular diseases [4,5].

Additionally, microbial contamination can exacerbate this imbalance by perpetuating inflammation through sustained immune cell activation and the release of pro-inflammatory cytokines, thereby elevating the risk of necrosis, biofilm formation, and recurrent infections that further complicate clinical management [6,7]. Pathogenic microorganisms commonly isolated from infected wounds include Escherichia coli, Pseudomonas aeruginosa, Proteus mirabilis, Staphylococcus aureus, Staphylococcus epidermidis, and Enterococcus faecalis, among others. Accordingly, infection prevention and/or treatment is a critical priority in chronic wound care to promote healing and prevent adverse outcomes [8].

Within this context, fibroblasts play a central role in wound healing, as they not only synthesize the extracellular matrix (ECM) but also contribute significantly to its remodeling. In addition, fibroblasts actively modulate the inflammatory response [9]. During healing, a subpopulation of fibroblasts differentiates into myofibroblasts—highly contractile cells that express α-smooth muscle actin (α-SMA) and contribute to wound contraction, collagen organization, and ECM remodeling [10,11,12]. However, in delayed-healing wounds, persistent activation of myofibroblasts may drive tissue fibrosis and stiffness, underscoring the importance of tightly regulating their differentiation and activity [4,13].

In recent years, there has been growing interest in the use of natural compounds with anti-inflammatory and antioxidant properties as potential therapeutic agents to enhance wound healing. Among these, ellagic acid and punicalagin—both found in the fruit of *Punica granatum* (pomegranate)—have demonstrated beneficial effects on health due to their favorable biological activities in various experimental models [14,15,16]. In particular, ellagic acid, a polyphenol with potent antioxidant properties, has been shown to reduce the production of reactive oxygen species (ROS) and modulate several inflammatory signaling pathways. Multiple studies have highlighted its capacity to inhibit inflammatory mediators, suggesting its potential in regulating wound healing processes in damaged tissues [17]. Similarly, punicalagin, an ellagitannin with antioxidant activity, has also been investigated for its ability to attenuate inflammatory responses in both animal and cellular models [18,19].

Despite these findings, evidence regarding their specific impact on human fibroblasts exposed to a pro-inflammatory environment remains limited. Given the pivotal role of fibroblasts in tissue repair, understanding whether these compounds modulate inflammation, cell proliferation, and migration under inflammatory conditions—as well as their potential antimicrobial activity—could support the development of novel strategies for managing chronic wounds. Therefore, the objective of this study was to evaluate the anti-inflammatory effects of ellagic acid and punicalagin in cultured human fibroblasts, assess their influence on cellular proliferation and migration in an inflammatory context, and explore their possible antimicrobial activity.

## 2. Results

### 2.1. Cell Proliferation Assay in the Presence of SIM, and LPS Medium

The results of the cell proliferation assay conducted under the Simulated Inflammatory Medium (SIM), which included Interleukin-1β (IL-1β), Interleukin-6 (IL-6), and Tumor Necrosis Factor-α (TNF), demonstrated a significant increase in the proliferation of human fibroblasts treated with either ellagic acid or punicalagin at both tested concentrations (10^−6^ M and 10^−7^ M), compared to the control group. Similarly, fibroblasts exposed to lipopolysaccharide (LPS, 10 ng/mL) and treated with the polyphenolic compounds also exhibited a significant increase in cell proliferation at all concentrations tested (Figure 1). The solvent control showed no impact on fibroblast proliferation.

### 2.2. Anti-Inflammatory Activity Assay

Quantification of IL-1β levels in the culture supernatant revealed a significant decrease in fibroblast cultures treated with ellagic acid or punicalagin at both concentrations (10^−6^ M and 10^−7^ M), compared to untreated controls. Regarding IL-6, a statistically significant reduction was observed only in the cultures treated with punicalagin at both concentrations (Figure 2). The solvent control did not influence cytokine secretion levels.

### 2.3. Cell Migration Assay in the Presence of SIM

Figure 3 presents the results of the migration capacity of human fibroblasts treated with ellagic acid or punicalagin at the selected concentrations in the presence of SIM, which contains IL-1β, IL-6, and TNF. A significant increase in wound closure percentage was observed at 24 h post-treatment with either compound, regardless of the dose, compared to untreated cells. No alterations in fibroblast migration were detected in the solvent control.

### 2.4. Antimicrobial Activity Assay

Quantification of IL-1β levels in the culture supernatant revealed a significant decrease in fibroblast cultures treated with ellagic acid or punicalagin at both concentrations (10^−6^ M and 10^−7^ M), compared to untreated controls. Regarding IL-6, a statistically significant reduction was observed only in the cultures treated with punicalagin at both concentrations (Figure 4). The positive controls (gentamicin and nystatin disks) produced clear inhibition zones, whereas the negative controls showed no antimicrobial effect.

## 3. Discussion

Wound healing is a dynamic and tightly regulated process involving the coordinated action of cytokines, growth factors, and proteolytic enzymes, among other mediators, to promote cellular regeneration and restore tissue integrity [20]. In acute wounds, inflammation plays a critical role by facilitating the clearance of external agents, such as pathogenic microorganisms, and stimulating the proliferation and migration of fibroblasts to the injury site [21]. These fibroblasts actively contribute to the synthesis of new ECM components and initiate re-epithelialization. However, a prolonged inflammatory response can disrupt the balance of the healing process, impairing the transition from the inflammatory to the proliferative phase. Such dysregulation may lead to the development of chronic wounds, characterized by compromised tissue repair and delayed resolution of tissue damage [22].

In this study, we demonstrated that both ellagic acid and punicalagin, phenolic compounds found in *Punica granatum*, promote fibroblast proliferation under inflammatory conditions and enhance the migratory capacity of cultured human fibroblasts. Moreover, these compounds significantly reduced the expression of cytokines associated with LPS-induced inflammation, suggesting their potential to accelerate tissue repair under inflammatory stress. These findings support the relevance of natural compounds as potential therapeutic agents for managing chronic wounds and dysregulated healing processes.

According to our results, ellagic acid and punicalagin sometimes exerted different effects at lower concentrations (10^−7^ M) compared to higher doses (10^−6^ M), in the assays performed. This effect is consistent with the biphasic or hormetic responses often described for phenolic compounds, in which nonlinear dose-dependent effects could be explained by the differential activation of redox and signaling pathways. Previous in vitro studies have reported similar results with pomegranate-derived compounds, supporting the nonlinear effects observed in our work. Previous assays conducted by our research group similarly showed that punicalagin and ellagic acid im-proved fibroblast viability and migration at concentrations of 10^−5^, 10^−6^, and 10^−7^ M, while punicalagin produced no changes in viability at 10^−9^ M, indicating a dose-dependent reversal of effects [23]. Similarly, Olchowik-Grabarek et al. [24] observed that enriched pomegranate extracts, which include ellagic acid and punicalagin, exhibited opposite actions depending on the cell type, protecting erythrocytes against oxidative stress while increasing ROS production and cell death in HeLa cells within the range of 25–100 µg/mL. In cancer cell lines (HeLa, HepG2), punicalagin and related extracts exerted cytotoxic or proapoptotic actions at higher concentrations, while they did not affect or affected non-tumor cells less severely [25,26]. The observed effects may be explained by the modulation of key cellular signaling pathways. Phenolic compounds such as those studied here may exert their effects through inhibition of the NF-κB (Factor Nuclear Kappa B) and MAPK signaling pathways, both of which are involved in the activation of genes associated with pro-inflammatory responses [27,28]. The NF-κB pathway is a central regulator of inflammation, controlling the transcription of pro-inflammatory genes including cytokines, chemokines, and adhesion molecules. Due to its pivotal role in inflammation, NF-κB has been extensively targeted in the development of anti-inflammatory therapies. However, recent genetic studies in murine models have revealed a dual role for NF-κB, demonstrating both pro- and anti-inflammatory effects, thus complicating its therapeutic targeting in chronic inflammatory conditions [29]. Within the inflammatory cascade, the expression of adhesion molecules such as E-selectin, ICAM-1, and VCAM-1 is highly dependent on NF-κB activation. These molecules are essential for leukocyte adhesion to the endothelium and subsequent extravasation into inflamed tissues [30]. In addition, microRNAs have emerged as key regulators of the inflammatory response by modulating adhesion molecule expression via two complementary mechanisms: indirectly by regulating NF-κB signaling, and directly by binding to the mRNA of adhesion molecules to fine-tune the inflammatory cascade [30]. Inhibition of NF-κB by ellagic acid and punicalagin may disrupt this positive feedback loop, thereby attenuating inflammation and creating a more favorable environment for tissue repair.

The MAPK pathway, which includes Extracellular signal-regulated kinases (ERK), Jun N-terminal kinases (JNK), and p38 kinases, also plays a critical role in cellular responses to inflammatory stress and regulates the production of pro-inflammatory cytokines [31]. These kinases are activated through dual phosphorylation on threonine and tyrosine residues in response to inflammatory cytokines and other environmental stressors. JNK and p38 pathways are predominantly activated by TNF and IL-1β and are involved in cell cycle arrest, DNA repair, or apoptosis, depending on the physiological context [32]. In this regard, the phenolic compounds examined may exert their modulatory effects by interfering with the activation of these signaling cascades, thereby regulating the production of inflammatory mediators at both the transcriptional and translational levels [33]. Such modulation would preserve fibroblast functionality, enabling key wound healing processes such as collagen synthesis and ECM remodeling, and thereby facilitating efficient tissue repair.

Regarding ellagic acid, multiple in vitro studies have demonstrated its anti-inflammatory properties [34,35]. For instance, Guan et al. [34] reported that treating RAW 264.7 murine macrophages with ellagic acid at 1, 2, and 4 μg/mL for 24 h prior to LPS stimulation led to a dose-dependent reduction in the pro-inflammatory cytokines TNF, IL-6, and IL-1β, along with an increase in Interleuquin-10 (IL-10), an anti-inflammatory cytokine. Similarly, BenSaad et al. [35] showed that ellagic acid at 50, 100, 150, and 200 μg/mL significantly decreased nitric oxide, prostaglandin E2, and IL-6 production in the same cell line, confirming its anti-inflammatory capacity. In vivo studies have also demonstrated the anti-inflammatory activity of ellagic acid. Gu et al. [36] evaluated its protective effects against LPS/D-galactosamine-induced acute liver injury in mice, administering intraperitoneal doses of 5, 10, and 20 mg/kg. The treatment significantly reduced TNF levels in both serum and liver tissue through NF-κB pathway inhibition [36]. Similarly, Liu et al. [37] showed that administering ellagic acid (100 mg/kg) for seven days attenuated LPS-induced neuroinflammation in rats. Treatment reversed LPS-induced increases in heme oxygenase-1, cyclooxygenase-2, and α-synuclein trimers in the substantia nigra, and decreased active caspase-3 and protein kinase-3 levels—biomarkers of neuronal damage and inflammation. These findings support the therapeutic potential of ellagic acid as an anti-inflammatory agent in LPS-induced models and suggest possible applications in various inflammatory pathologies. Consistent with this, our results corroborate the anti-inflammatory effects of ellagic acid in fibroblasts under inflammatory conditions, evidenced by the reduction in IL-1β levels.

The anti-inflammatory potential of punicalagin has also been demonstrated in various cell types. Cao et al. [38] found that punicalagin at 25 and 50 μM, administered one hour before LPS exposure, significantly attenuated the inflammatory response in RAW264.7 macrophages. This effect was associated with the inhibition of NF-κB and MAPK signaling, evidenced by reduced phosphorylation of p65, p38, JNK, and ERK, and the downregulation of the FoxO3a/autophagy pathway involved in cellular homeostasis. Olajide et al. [39] similarly showed that punicalagin significantly reduced TNF and IL-6 production in LPS-activated primary microglia. Subsequent studies demonstrated that punicalagin at 50 μM prevented memory impairment associated with neuroinflammation by inhibiting NF-κB activation and reducing oxidative stress in LPS-treated astrocytes and microglia. In vivo, the same authors administered punicalagin (1.5 mg/kg in water) concomitantly with LPS over four weeks to induce cerebral inflammation in mice. This treatment not only improved cognitive performance (as assessed by the Morris Water Maze and passive avoidance tests) but also significantly reduced TNF, IL-6, and IL-1β levels in brain tissue, highlighting the potential of punicalagin as a therapeutic agent in neuroinflammatory conditions [40]. Likewise, in our study, punicalagin significantly decreased IL-1β and IL-6 levels.

Reducing pro-inflammatory cytokines is particularly relevant in chronic wounds, where persistent inflammation can inhibit fibroblast migration, proliferation, and ECM synthesis—key processes for tissue regeneration [41,42]. In chronic wounds such as diabetic or pressure ulcers, the inflammatory microenvironment not only impairs healing but may also promote tissue degradation via upregulated Matrix Metalloproteinases (MMPs) [43]. Several studies have shown that phenolic compounds like hydroxytyrosol, tyrosol, and oleocanthal—found in extra virgin olive oil—can downregulate MMP gene expression in cultured human fibroblasts [44]. Additionally, both ellagic acid and punicalagin exhibit antioxidant properties by reducing ROS production and oxidative stress in various cell types and animal models. Specifically, ellagic acid has been shown to alleviate oxidative stress in piglets and mice via Nrf2 pathway activation, enhancing the expression of antioxidant enzymes [45,46]. Similarly, punicalagin effectively reduced ROS and oxidative stress in human keratinocytes, retinal pigment epithelial cells, and in a rodent model of ankylosing spondylitis [47]. These findings suggest that both compounds not only protect cellular functions but also enhance antioxidant enzyme activity, making them promising candidates for mitigating oxidative damage in chronic wounds and other pathologies.

Despite previous reports of antimicrobial properties [48,49], our findings did not show significant antimicrobial activity of ellagic acid or punicalagin against the tested microorganisms. However, this is consistent with other studies using these compounds against *Staphylococcus aureus* [50]. These discrepancies may be attributed to various factors. First, the concentrations used in cell cultures may not have been sufficient to elicit a direct antimicrobial effect, despite being effective in modulating inflammation. Additionally, media composition may have interfered with compound activity, partially neutralizing their antimicrobial potential. Alternatively, ellagic acid and punicalagin may exert indirect antimicrobial effects, as previously suggested, potentially through modulation of cellular immunity or alteration of the microenvironment that supports bacterial growth [51,52]. Therefore, the absence of direct antimicrobial activity in this model does not preclude their utility in wound management, as reducing inflammation may help prevent bacterial colonization, particularly in chronic wounds associated with biofilm formation.

The combined anti-inflammatory properties and the ability to promote fibroblast growth and migration under inflammatory conditions highlight the therapeutic potential of ellagic acid and punicalagin in wound healing. Controlling inflammation is also critical for preventing secondary infections and fostering a microenvironment conducive to tissue repair.

To advance the application of ellagic acid and punicalagin, further in vivo studies are essential to evaluate their efficacy in animal wound models and determine whether the in vitro effects translate into clinically meaningful outcomes. Additionally, exploring combination therapies with antimicrobial agents could enhance infection control in complex wounds. Finally, the development of topical formulations such as gels, dressings, or creams for controlled release of the active compounds may optimize their anti-inflammatory properties and facilitate their therapeutic use.

## 4. Materials and Methods

### 4.1. Preparation and Dilution of Treatment Solutions

Ellagic acid (≥95% purity, HPLC; Sigma-Aldrich, St. Louis, MO, USA) purified, and punicalagin (≥98% purity, HPLC; Sigma-Aldrich) purified both from pomegranate were used for the in vitro assays (Figure 5). Stock solutions of both polyphenols were prepared at a concentration of 10^−3^ M. Punicalagin was initially dissolved in methanol (5 mg/mL) and subsequently diluted with Milli-Q water (Millipore Corp., Bedford, MA, USA). Ellagic acid was solubilized in 1 M NaOH (10 mg/mL) and then adjusted with Milli-Q water. All stock solutions were stored at –20 °C until use. Solvent controls were included in all experiments, treating fibroblasts with the same concentrations of methanol (for punicalagin) or NaOH (for ellagic acid) used to prepare the stock solutions.

Prior to experimentation, the stock solutions were diluted to final concentrations of 10^−6^ M and 10^−7^ M in Dulbecco’s Modified Eagle Medium (DMEM; Gibco, Carlsbad, CA, USA) supplemented with 10% Fetal Bovine Serum (FBS; Sigma-Aldrich). The selection of these concentrations was based on previous studies conducted by our group, in which a dose–response viability assay indicated that 10^−6^ M and 10^−7^ M optimized cell viability without inducing cytotoxicity in cultured human fibroblasts [23].

### 4.2. Cell Culture

The human dermal fibroblast cell line CCD-1064Sk (Ref: CRL-2076) was obtained from the American Type Culture Collection (ATCC, Manassas, VA, USA) through the Center for Scientific Instrumentation (University of Granada, Granada, Spain). Cells were cultured in T25 flasks (BD Falcon™, Becton Dickinson Labware, Franklin Lakes, NJ, USA) in DMEM supplemented with 10% FBS, 1% L-glutamine (Sigma-Aldrich), 2% HEPES (Sigma-Aldrich), 2.5 µg/mL amphotericin B (Sigma-Aldrich), 100 IU/mL penicillin (ERN Laboratories, Barcelona, Spain), and 50 µg/mL gentamicin (B. Braun Medical S.A., Jaén, Spain). Cultures were maintained at 37 °C in a humidified atmosphere with 5% CO_2_, and the medium was replaced every three days.

Upon reaching approximately 80% confluence, cells were detached using 0.05% trypsin and 0.02% EDTA (both from Sigma-Aldrich), washed with Phosphate-Buffered Saline (PBS; Sigma-Aldrich), and resuspended in fresh medium containing 10% FBS for further subculturing.

### 4.3. Cell Proliferation Assay

Fibroblast proliferation was assessed using the MTT colorimetric assay (Sigma-Aldrich), which quantifies the reduction in the yellow tetrazolium salt MTT (3-(4,5-dimethylthiazol-2-yl)-2,5-diphenyltetrazolium bromide) into purple formazan crystals by metabolically active cells. This metabolic conversion is directly proportional to the number of viable cells.

Cells were seeded at a density of 2 × 10^4^ cells/mL in 96-well plates (Falcon™) with DMEM supplemented with 10% FBS. To simulate an inflammatory environment, cultures were exposed for 24 h to a SIM containing IL-1β (100 pg/mL), IL-6 (6000 pg/mL), and TNF (28 pg/mL) (Sigma-Aldrich). These cytokine concentrations were selected based on previous studies modeling inflammatory microenvironments and are within the range described in chronic wound fluids [42,53,54,55]. After 24 h, SIM was replaced with fresh medium containing the same cytokines, along with ellagic acid or punicalagin at 10^−6^ M or 10^−7^ M.

In parallel, following Díez-Tercero et al. [56], additional fibroblast cultures were seeded into separate 96-well plates and treated with DMEM supplemented with LPS (Sigma-Aldrich) at 10 ng/mL to mimic a Gram-negative bacterial inflammatory response. Under these conditions, the same polyphenol concentrations were evaluated over a 24 h period.

After treatment, culture media were replaced with phenol red-free DMEM containing MTT (5 mg/mL). Following 4 h of incubation at 37 °C, formazan crystals were dissolved with Dimethyl Sulfoxide (DMSO; Sigma-Aldrich), and absorbance was measured at 570 nm using a Sunrise™ microplate reader (TECAN, Männedorf, Switzerland). The results were used to determine changes in fibroblast proliferation under cytokine- and LPS-induced inflammatory conditions.

Control groups were established to ensure accurate comparisons. For cytokine and LPS treatments, the respective control wells contained the same inflammatory stimuli without polyphenol treatments. An additional control group for the LPS condition received only DMSO, to control for solvent effects.

### 4.4. Analysis of Pro-Inflammatory Cytokine Production

To assess the anti-inflammatory activity of ellagic acid and punicalagin under LPS-induced inflammatory conditions, fibroblasts were seeded at a density of 2 × 10^4^ cells/mL in 96-well plates. Once adhered, cells were exposed to 10 ng/mL LPS in the presence or absence of ellagic acid or punicalagin (10^−6^ M and 10^−7^ M) for 48 h.

Following incubation, supernatants were collected, and levels of IL-1β and IL-6 were quantified using commercial Enzyme-Linked Immunosorbent Assay (ELISA) kits (Abcam; IL-1β: ab46052; IL-6: ab178013), according to the manufacturer’s protocols. Cytokine concentrations were expressed in picograms per milliliter (pg/mL).

### 4.5. Cell Migration Assay

Cell migration was evaluated using the culture insert method, as described by Cappiello et al. [57]. Human fibroblasts were suspended in DMEM containing IL-1β, IL-6, and TNF at the same concentrations used in the proliferation assay, at a final cell density of 4 × 10^5^ cells/mL. A 70 µL aliquot of the cell suspension was added to each well of a two-chamber culture insert (Ibidi, Munich, Germany) placed in a 24-well plate. After 24 h of incubation, the inserts were gently removed, creating a uniform cell-free gap (~200 µm wide). Wells were rinsed with PBS to remove detached cells, and ellagic acid or punicalagin (10^−6^ M or 10^−7^ M) were added in cytokine-supplemented DMEM.

Plates were returned to standard culture conditions, and images of the wound area were captured at 0, 12, and 24 h post-treatment using an inverted phase-contrast microscope. Gap closure was quantified using Motic Images Plus 3.0 software (Motic China Group Co., Hong Kong, China; Available online: https://www.motic.com/, accessed on 4 September 2025), and the percentage of wound closure was calculated as:*Wound closure* (%) = (*W*_0_ − *W*_*n*_)/*W*_0_ × 100%;
where *W*_0_ is the initial width of the space immediately after the culture insert assay, and *Wn* is the width at the different measurement time points.

### 4.6. Antimicrobial Capacity Assay

The antimicrobial properties of ellagic acid and punicalagin were assessed using the Kirby–Bauer disk diffusion method. The tested strains included *Candida albicans*, *Staphylococcus aureus*, and *Pseudomonas aeruginosa*. Microorganisms were cultured on Mueller–Hinton agar (MHA; Sigma-Aldrich) for 24 h at 37 °C. Cell suspensions were then prepared in Tryptic Soy Broth (TSB; Becton Dickinson) at 10^8^ CFU/mL (0.5 McFarland standard; OD_600_ ≈ 0.1).

Using sterile cotton swabs (Sigma-Aldrich), the microbial suspensions were evenly spread over MHA plates, which were air-dried for 30 min under laminar flow. Sterile cellulose disks (5 mm diameter) impregnated with 10 µL of ellagic acid or punicalagin (10^−6^ M or 10^−7^ M) were placed on the agar surface. Plates were incubated at 37 °C for 24 h, and the diameter of the inhibition zones was measured using a standard ruler. All tests were performed in triplicate.

Positive controls consisted of commercial antibiotic and antifungal disks (gentamicin, Sigma-Aldrich, Cat. No. G1272 for bacteria; and nystatin, Sigma-Aldrich, Cat. No. N3503 for *Candida albicans*). Likewise, negative controls consisted of disks loaded with the same solvent used to prepare the test compounds.

### 4.7. Statistical Analysis

Data are presented as mean ± standard error of the mean (SEM). Normality was assessed using the Shapiro–Wilk test. For normally distributed variables, comparisons between two independent groups were performed using Student’s t-test; otherwise, the Mann–Whitney U test was applied. All statistical analyses were conducted using SPSS v29 (IBM Corp., Armonk, NY, USA), with statistical significance set at *p* ≤ 0.05. At least three independent experiments were conducted for each assay.

## 5. Conclusions

Both ellagic acid and punicalagin enhance the proliferation and promote the migratory capacity of cultured human fibroblasts under inflammatory conditions. In addition, they significantly reduce the synthesis of cytokines associated with LPS-induced inflammation. These properties confer upon both compounds the potential to accelerate tissue regeneration in the context of inflammatory stress. Although neither compound exhibited direct antimicrobial activity in the tested model, our findings suggest that ellagic acid and punicalagin may play a critical role in modulating the inflammatory response and, consequently, in improving wound healing. These results lay the groundwork for future investigations aimed at optimizing their therapeutic application and evaluating their impact in more complex biological contexts.

## Figures and Tables

**Figure 1 ijms-26-08681-f001:**
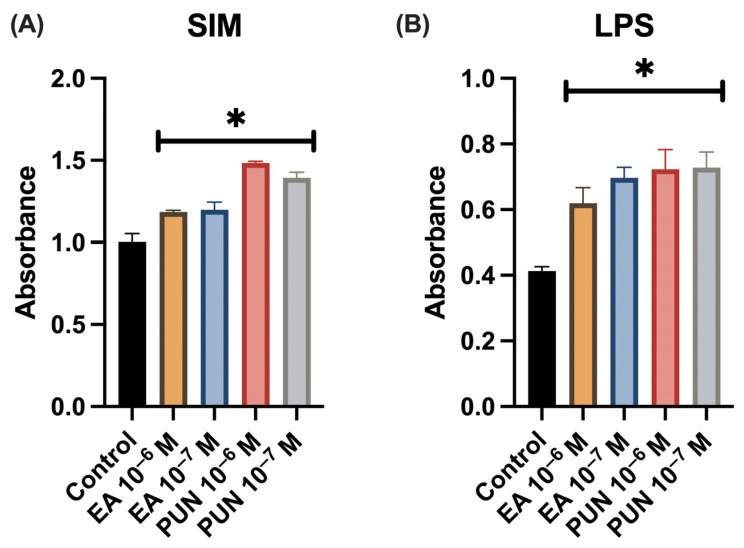
Cell proliferation in human fibroblasts treated with ellagic acid (EA) or punicalagin (PUN) at concentrations of 10^−6^ M and 10^−7^ M in the presence of (**A**) Simulated Inflammatory Medium (SIM; IL-1β, IL-6, TNF) or (**B**) LPS. Data are expressed as absorbance values relative to untreated control samples (* *p* ≤ 0.05).

**Figure 2 ijms-26-08681-f002:**
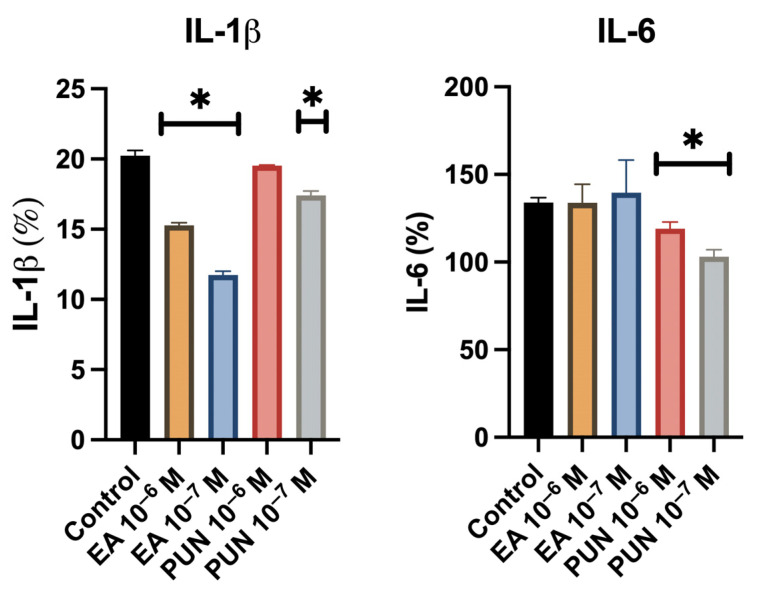
IL-1β and IL-6 levels in the supernatants of human fibroblast cultures treated with ellagic acid (EA) or punicalagin (PUN) at concentrations of 10^−6^ M and 10^−7^ M in the presence of LPS (10 ng/mL). Data are expressed as a percentage relative to untreated control samples (* *p* ≤ 0.05).

**Figure 3 ijms-26-08681-f003:**
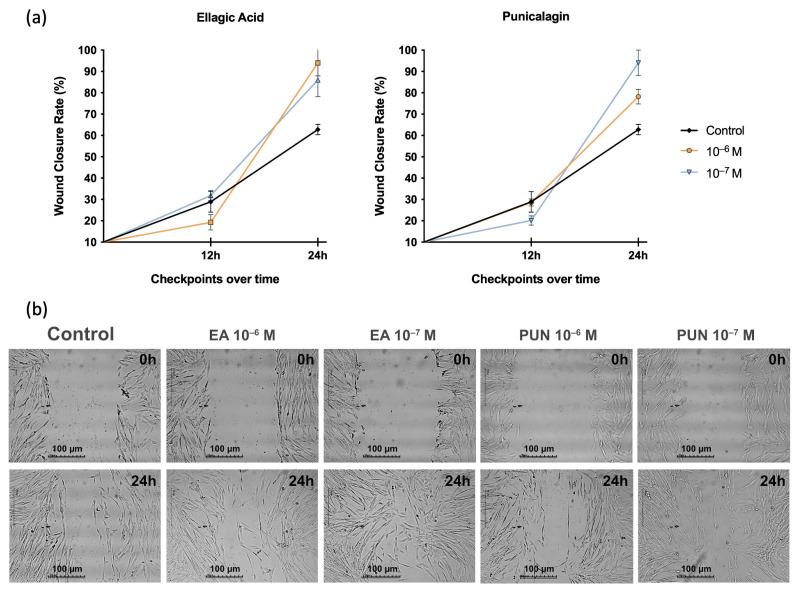
(**a**) Effect of ellagic acid (EA) and punicalagin (PUN) on the migratory capacity of human fibroblasts in the presence of SIM (medium supplemented with IL-1β, IL-6, and TNF). EA and PUN at 10^−6^ M and 10^−7^ M significantly enhanced fibroblast migration at 24 h (*p* = 0.009, *p* = 0.046, *p* = 0.021, and *p* = 0.008, respectively). (**b**) Brightfield microscopy images at 0 and 24 h post-treatment demonstrate a marked reduction in the wound area in all treated groups compared to control cells. This enhancement in wound closure was evident across all concentrations of EA and PUN evaluated.

**Figure 4 ijms-26-08681-f004:**
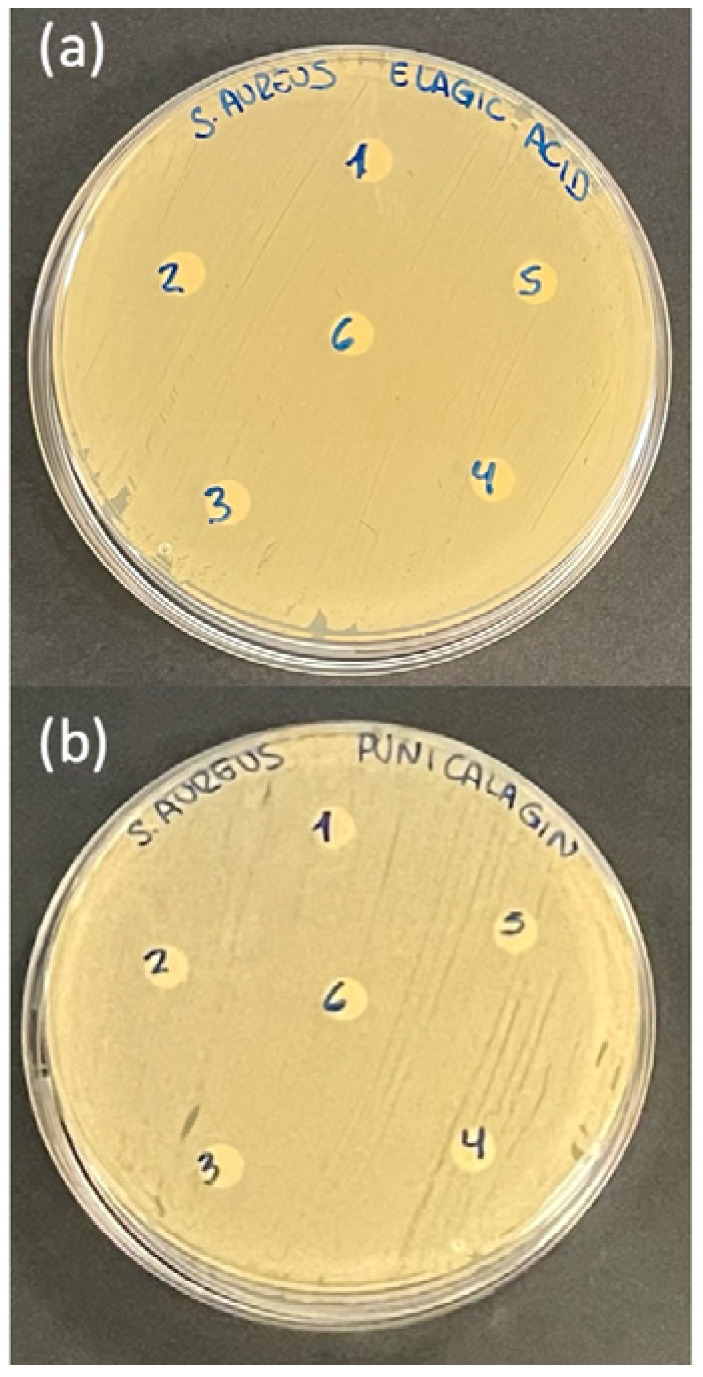
Cultures of *Staphylococcus aureus* treated with (**a**) ellagic acid (EA) and (**b**) punicalagin (PUN) at concentrations of 10^−6^ M and 10^−7^ M. No inhibition zones were observed, indicating the absence of antimicrobial activity.

**Figure 5 ijms-26-08681-f005:**
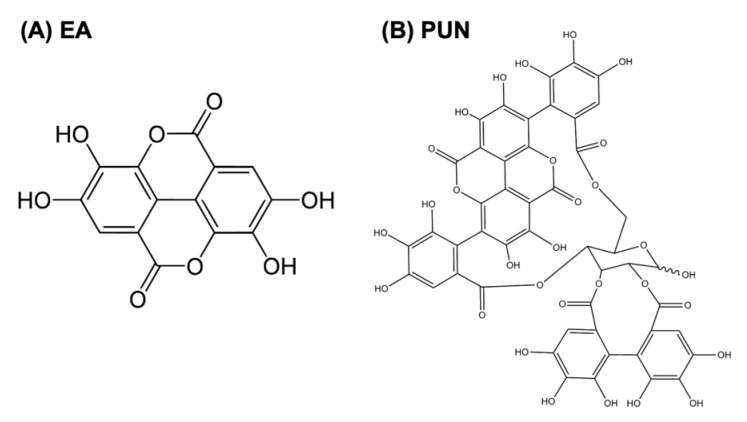
Chemical structures of (**A**) ellagic acid (EA) and (**B**) punicalagin (PUN). Both polyphenols are naturally occurring compounds derived from Punica granatum (pomegranate).

## Data Availability

Datasets generated during and/or analyzed during the current study are available from the corresponding author upon request.

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
