# Peer review of "Anti-Inflammatory and Antimicrobial Effect of Ellagic Acid and Punicalagin in Dermal Fibroblasts"

_ijms, 2025, doi:10.3390/ijms26178681_

Round 1

Reviewer 1 Report

Comments and Suggestions for Authors

The authors showed the potential of ellagic acid and punicalagin as Anti-Inflammatory in dermal fibroblasts. Both compounds significantly enhanced fibroblast viability and migration under inflammatory conditions and reduced the secretion of IL-1β and IL-6.

I recommend this article for publication after addressing the following concerns.

  1. Line 20 ……a simulated inflammatory microenvironment induced by proinflammatory cytokines (IL- 1β, IL-6, TNF-α)… author must provide the full name before using this abbreviation.
  2. Author must include the chemical structures of the ellagic acid and punicalagin.
  3. Author showed the cell proliferation of fibroblasts under SIM conditions. Any explanation for why the cell proliferation was less at 10-7M as compared to 10-6 M dose treatments for PUN.
  4. Similarly, the measured IL-1β and IL-6 levels in the supernatants of human fibroblast cultures treated with ellagic acid (EA) or punicalagin (PUN) at concentrations of 10⁻⁶ M and 10⁻⁷ M in the presence of LPS showed the more suppression of IL-1β and IL-6 in case of EA 10-7M and PUN 10-7M, respectively as compared to higher dose of EA and PUN ( i.e 10⁻⁶ M ). Same was observed for migration studies of fibroblasts where PUN was much more effective at lower concentration than the higher concentration.
  5. The ellagic acid and punicalagin compounds did not showed any antimicrobial activities at the tested concentration. Did author has tested these compounds at any higher concentrations for antimicrobial activities in Staphylococcus aureus if not please check the result at higher concentrations? Or any other microbes had been tested?

I recommend for publication after this minor revision….

Author Response

Comment 1: Line 20……a simulated inflammatory microenvironment induced by proinflammatory cytokines (IL- 1β, IL-6, TNF-α)… author must provide the full name before using this abbreviation.

Response 1:

Thank you for this observation. In the revised manuscript, we have provided the full names of the cytokines before their abbreviations (line 20): “…interleukin-1β (IL-1β), interleukin-6 (IL-6), and tumour necrosis factor-α (TNF)…”.

In addition, the abstract has been briefly modified to comply with the journal’s 200-word limit.

Comments 2: Author must include the chemical structures of the ellagic acid and punicalagin.

Response 2:

We appreciate this suggestion. The chemical structures of ellagic acid and punicalagin have been added as Figure 5 in the Materials and Methods section (‘4.1. Preparation and Dilution of Treatment Solutions’) to enhance clarity of the compounds used in this study.

Comment 3: Author showed the cell proliferation of fibroblasts under SIM conditions. Any explanation for why the cell proliferation was less at 10-7M as compared to 10-6 M dose treatments for PUN.

Response 3:

We agree that proliferation was slightly lower than 10⁻⁷ M compared to 10⁻⁶ M. Although the study showed that both compounds significantly increased the viability and migration of fibroblasts under inflammatory conditions, it is true that some natural compounds may exhibit a nonlinear or biphasic concentration-dependent response that is frequently observed in polyphenolic compounds, potentially acting through hormetic mechanisms, in which lower concentrations can sometimes cause attenuated effects compared to higher concentrations. We have added a paragraph in the Discussion section to clarify this point and have referenced the relevant literature on the biphasic dose-response effects of these polyphenols (line 171).

Comment 4: Similarly, the measured IL-1β and IL-6 levels in the supernatants of human fibroblast cultures treated with ellagic acid (EA) or punicalagin (PUN) at concentrations of 10⁻⁶ M and 10⁻⁷ M in the presence of LPS showed the more suppression of IL-1β and IL-6 in case of EA 10-7M and PUN 10-7M, respectively as compared to higher dose of EA and PUN ( i.e 10⁻⁶ M ). Same was observed for migration studies of fibroblasts where PUN was much more effective at lower concentration than the higher concentration.

Response 4:

Thank you for this comment. We agree this apparent paradox deserves clarification. Studies have reported results consistent with the phenomenon of hormesis or biphasic responses, in which at low/moderate doses, some phenols act as signaling molecules/inducers of protective and repair pathways, while at higher doses they can behave as pro-oxidants or activate apoptosis in sensitive cells, which supports the nonlinear effects observed in our work. We have expanded the Discussion section to highlight this aspect and provide references supporting this phenomenon (line 171).

Comment 5: The ellagic acid and punicalagin compounds did not showed any antimicrobial activities at the tested concentration. Did author have tested these compounds at any higher concentrations for antimicrobial activities in Staphylococcus aureus if not please check the result at higher concentrations? Or any other microbes had been tested?

Response 5:

We sincerely appreciate your valuable suggestion and the opportunity to clarify this point. In preliminary work, we tested both compounds at different concentrations up to 10⁻² M against Candida albicans, Staphylococcus aureus, and Pseudomonas aeruginosa. No zones of inhibition were observed in any case. These results confirmed that ellagic acid and punicalagin lacked direct antimicrobial activity even at much higher concentrations, so we chose to use only the lowest non-cytotoxic concentrations (10⁻⁶ and 10⁻⁷ M) in the assays presented in the manuscript

Reviewer 2 Report

Comments and Suggestions for Authors

This is an interesting submission focused on testing the anti-inflammatory and anti-microbial impacts of ellagic acid and punicalagin, substances found in pomegranates, in the context of abnormal skin wound healing. Unfortunately, some aspects of the experiments were difficult to assess in their current form, and the following questions are designed to further enhance the clarity of the submission.

  • How many experimental replicates were performed for each of the assays reported?
  • For each of the experiments, were standard deviations (SD) or stand errors (SE) reported?
  • Were vehicle controls (i.e., solvent controls - the same methanol or NaOH concentrations used to dilute the stock solutions of ellagic acid and punicalagin) utilised for these analyses? If not, how can the authors distinguish any impacts of the solvents from those derived from the molecules being tested?
  • What was the rationale for the cytokine concentrations used in the SIMs? Previous studies have justified the concentrations of the individual components of SIMs (also known as “cytomixes” e.g., Close TE et al, Microcirculation. 2013 Aug;20(6):534-43. doi: 10.1111/micc.12053. PMID: 23441883, and Zhang W et al, Basic Res Cardiol. 2017 Mar;112(2):16. doi: 10.1007/s00395-017-0603-8. Epub 2017 Feb 6. PMID: 28168403) by comparisons to serum concentrations in sepsis. Are the concentrations of cytokines used by the authors in this study reflective of localized levels in skin wounds? If so, can references supporting these concentrations be provided?
  • Were no positive controls included in the Kirby-Bauer disk diffusion assays? The images do not indicate any inhibition zones at all; can the authors confirm that ellagic acid and punicalagin are capable of diffusing from a disk into agar? Were vehicle controls (solvents) utilised for these analyses?
  • Figure 3a: Why was there no 12hr assessments for punicalagin at 10-6 M? The linear result reported seems very unlikely, considering the clearly bi-phasic effects at 10-7 M.

Minor edits: One suggested but not compulsory edit: the TNF nomenclature has been subject to much debate since the renaming of TNF-b to Lymphotoxin-a more than 20 years ago. The authors could reconsider the use of outdated TNF-a nomenclature and replace it with the more modern and, arguably more appropriate, TNF nomenclature.

Author Response

Comment 1: How many experimental replicates were performed for each of the assays reported?

Response 1: Thank you for this comment. Regarding the tests, we confirm that all tests were performed at least in triplicate. We have clarified this information in the Materials and Methods section (‘4.7. Statistical analysis’).

Comment 2: For each of the experiments, were standard deviations (SD) or stand errors (SE) reported?

Response 2: We confirm that the results of all experiments are presented as mean ± standard error of the mean (SEM), in accordance with the statistical analysis section of our manuscript.

Comment 3: Were vehicle controls (i.e., solvent controls - the same methanol or NaOH concentrations used to dilute the stock solutions of ellagic acid and punicalagin) utilised for these analyses? If not, how can the authors distinguish any impacts of the solvents from those derived from the molecules being tested?

Response 3: We appreciate your comment. Vehicle controls were indeed included in all experiments. Specifically, fibroblasts were treated with the same concentrations of methanol (solvent for punicalagin) or NaOH (solvent for ellagic acid) used to dilute the compounds. These controls showed no significant effects on cell viability, migration, or cytokine secretion, confirming that the observed effects were due to the polyphenols. We have added this clarification to the Materials and Methods section (‘4.1. Preparation and Dilution of Treatment Solutions’) and further clarified the corresponding control results in the Results section.

Comment 4: What was the rationale for the cytokine concentrations used in the SIMs? Previous studies have justified the concentrations of the individual components of SIMs (also known as “cytomixes” e.g., Close TE et al, Microcirculation. 2013 Aug;20(6):534-43. doi: 10.1111/micc.12053. PMID: 23441883, and Zhang W et al, Basic Res Cardiol. 2017 Mar;112(2):16. doi: 10.1007/s00395-017-0603-8. Epub 2017 Feb 6. PMID: 28168403) by comparisons to serum concentrations in sepsis. Are the concentrations of cytokines used by the authors in this study reflective of localized levels in skin wounds? If so, can references supporting these concentrations be provided?

Response 4:

Thank you for your comment. The cytokine concentrations used (IL-1β: 100 pg/mL, IL-6: 6000 pg/mL, TNF: 28 pg/mL) were selected based on levels reported in chronic wound fluids rather than in serum. While cytokine concentrations vary considerably depending on the type and stage of the wound, these values fall within the ranges documented in wound exudates. To clarify this point, we have added this information to the Materials and Methods section (‘4.3. Cell proliferation assay’), along with references to previous studies that reported comparable concentrations in chronic wound fluids.

Comment 5: Were no positive controls included in the Kirby-Bauer disk diffusion assays? The images do not indicate any inhibition zones at all; can the authors confirm that ellagic acid and punicalagin are capable of diffusing from a disk into agar? Were vehicle controls (solvents) utilised for these analyses?

Response 5: We appreciate your comment regarding the use of controls in Kirby-Bauer disk diffusion assays. Positive controls (gentamicin or nystatin discs) and negative controls (discs loaded with the same solvent used for the treatments) were included in all experiments. The positive controls showed clear zones of inhibition, confirming the validity of the assay, while the negative controls showed no antimicrobial effect. We have included this information in the Materials and Methods section (‘4.6. Antimicrobial Capacity Assay’) and have also described the corresponding results in the Results section.

Comment 6: Figure 3a: Why was there no 12hr assessments for punicalagin at 10-6 M? The linear result reported seems very unlikely, considering the clearly bi-phasic effects at 10-7 M.

Response 6: Thank you for pointing out the error in Figure 3a. We apologize for the omission. Data were collected at 0, 12, and 24 hours for all groups, including the 10⁻⁶ M punicalagin group. The 12-hour values showed an intermediate effect consistent with the overall trend. We have updated the figure 3a to include this missing data.

Comment 7: Minor edits: One suggested but not compulsory edit: the TNF nomenclature has been subject to much debate since the renaming of TNF-b to Lymphotoxin-a more than 20 years ago. The authors could reconsider the use of outdated TNF-a nomenclature and replace it with the more modern and, arguably more appropriate, TNF nomenclature.

Response 7:

We thank the reviewer for this note. We have revised the manuscript to replace TNF-α with TNF where appropriate, in accordance with current nomenclature.

Round 2

Reviewer 2 Report

Comments and Suggestions for Authors

Substantially improved, no further edits